# Should Carbohydrate Intake Be More Liberal during Oral and Enteral Nutrition in Type 2 Diabetic Patients?

**DOI:** 10.3390/nu15020439

**Published:** 2023-01-14

**Authors:** Ondrej Sobotka, Marie Ticha, Marketa Kubickova, Petr Adamek, Lenka Polakova, Vojtech Mezera, Lubos Sobotka

**Affiliations:** 13rd Department of Medicine, Metabolic Care and Gerontology, Medical Faculty, Charles University, 50005 Hradec Kralove, Czech Republic; 2Levit’s Aftercare Centre, 50801 Horice, Czech Republic; 3Geriatric Center, Pardubice Hospital, 53203 Pardubice, Czech Republic

**Keywords:** enteral nutrition, carbohydrate intake, diabetes mellitus, insulin resistance, glucose, glucose metabolism

## Abstract

Carbohydrate (CHO) intake in oral and enteral nutrition is regularly reduced in nutritional support of older patients due to the high prevalence of diabetes (usually type 2—T2DM) in this age group. However, CHO shortage can lead to the lack of building blocks necessary for tissue regeneration and other anabolic processes. Moreover, low CHO intake decreases CHO oxidation and can increase insulin resistance. The aim of our current study was to determine the extent to which an increased intake of a rapidly digestible carbohydrate—maltodextrin—affects blood glucose levels monitored continuously for one week in patients with and without T2DM. Twenty-one patients (14 T2DM and seven without diabetes) were studied for two weeks. During the first week, patients with T2DM received standard diabetic nutrition (250 g CHO per day) and patients without diabetes received a standard diet (350 g of CHO per day). During the second week, the daily CHO intake was increased to 400 in T2DM and 500 g in nondiabetic patients by addition of 150 g maltodextrin divided into three equal doses of 50 g and given immediately after the main meal. Plasma glucose level was monitored continually with the help of a subcutaneous sensor during both weeks. The increased CHO intake led to transient postprandial increase of glucose levels in T2DM patients. This rise was more manifest during the first three days of CHO intake, and then the postprandial peak hyperglycemia was blunted. During the night’s fasting period, the glucose levels were not influenced by maltodextrin. Supplementation of additional CHO did not influence the percentual range of high glucose level and decreased a risk of hypoglycaemia. No change in T2DM treatment was indicated. The results confirm our assumption that increased CHO intake as an alternative to CHO restriction in type 2 diabetic patients during oral and enteral nutritional support is safe.

## 1. Introduction

Any acute or chronic disease, trauma, or surgery is associated with loss of appetite due to inflammation that stimulates a catabolic response in the body with a consequential loss of skeletal muscle mass that can result in sarcopenia [1]. After the systemic inflammation subsides, an important therapeutic goal is the regeneration of the affected tissues together with the adjustment of organ functions, the recovery of muscle mass, and the improvement of muscle function. This is especially true for older patients who lose muscle mass and strength very quickly, with a subsequent loss of independence [2]. Proper nutritional support, including enteral nutritional products, is required to prevent this process. Both special nutritional supplements and enteral tube feeding together with exercise are important in this regard [3,4]. At present, the main emphasis is placed on the protein component of nutrition and adequate total energy intake [5]. However, very little attention is paid to the carbohydrate (CHO) component of enteral nutrition. This is related to the fact that glucose and other CHOs in the diet are considered by most nutritionists and physicians to be only “empty” energy, without any further value for the organism. In addition, due to the increasing incidence of obesity and type 2 diabetes (T2DM) in the population, the current nutritional recommendation is to reduce the amount of carbohydrates in the diet of the general population [6,7]. In addition, an increase in the level of glucose in the blood due to insulin resistance during the progression of an acute disease is often incorrectly considered as an indication to reduce carbohydrate intake [6].

Currently, it is recommended to cover 50–55% of energy from CHOs and 30–35% of energy from fat in enteral nutrition products. This means that if patients are fed exclusively by enteral nutrition, they usually receive 250–270 g of CHOs, 75–100 g of protein and 65–77 g of lipids providing a total of 2000 kcal per day. In patients with type 2 diabetes who receive a special diabetic enteral diet, the intake of CHOs is even lower [8]. If patients need more energy supply, this is usually given in the form of fat. This is justified by nutritionists with the argument of the risk of hyperglycemia and potential T2DM development.

CHOs, which are consumed in food, are the main source of glucose for the body. Glucose, however, is not only a mere source of energy, but is also primarily an important metabolic substrate that plays an indispensable role for many metabolic events, including anabolic and reduction processes [9]. For this reason, increased CHO intake along with reduced fat intake could be beneficial for patients whose primary nutritional goal is to induce regeneration and an anabolic state, including increased muscle mass formation [9,10].

Because the incidence of T2DM increases with age, there is often concern that increased CHO intake may impair diabetes compensation in both T2DM and nondiabetic patients [11]. For this reason, T2DM, and patients with hyperglycemia and insulin resistance receive even smaller amounts of CHO both in the diet and in the supplementary nutritional products [12]. In this way, however, patients do not receive the sufficient glucose for anabolic processes and must rely on endogenous glucose production. In addition, an increased percentage of energy from dietary fat intake leads to increased fat accumulation in adipose tissue. Interestingly, the effect of increased carbohydrate intake on plasma glucose concentration and diabetes compensation in older adults with type 2 diabetes is not completely understood.

A higher anabolic effect is especially important for patients in whom we expect wound healing or an anabolic course of rehabilitation treatment. The result of the study could be useful for the development of new formulas and products for enteral nutrition and tube feeding, with the aim of increasing their anabolic potential. Therefore, the aim of our current study was to determine to what extent an increased intake of a rapidly digestible oligosaccharide/polysaccharide maltodextrin (containing 25–30 glucose units) will affect blood glucose levels.

## 2. Materials and Methods

The study was organized in the cooperation with Levit’s Aftercare Centre in Horice, Czech Republic, and was approved by the ethical committee of Faculty Hospital in Hradec Kralove No:202111P08. Levit’s Aftercare Centre in Horice is a long-term hospitalization facility specializing in treatment of patients with chronic wounds or elderly patients in need of a long-term rehabilitation and nutritional care who recently suffered and survived acute illness. Twenty-two patients (14 with type 2 diabetes and eight without diabetes) were included in this prospective cross-over study. Patients were hospitalized in order to improve their general condition after an acute illness or to heal chronic wounds. Physical rehabilitation and nutritional care were the main methods of treatment, and 17 patients underwent local treatment of chronic wounds. Inclusion criteria were as follows: sufficient oral food intake, stable condition of patients without any acute health problems (i.e., fever, vomiting, severe pain, unconsciousness, decompensated diabetes), and signed informed consent. Exclusion criteria were malnutrition with need of artificial nutrition supplements, instability due to acute health issues (see above), terminal stage of disease in and palliative care and refusal to enroll or continue in the study. Brief characterization of patients included in our study is depicted in Appendix A. During the first of two weeks of this longitudinal crossover study, patients received either a standard institutional diet (patients without diabetes) or a low-carbohydrate diet (patients with type 2 diabetes). The composition of both diets was according to the standard protocols at the Levit’s Aftercare Centre based on recommendations for hospital dietary system in rehabilitation centers in Czech Republic [13]. Maximum daily carbohydrate intake in the standard diet for nondiabetics was 350 g (57% of total energy intake) and 250 g (44% of total energy intake) for T2DM patients.

Food intake was monitored by the nursing staff using the Quarter–Waste method [3,14]. During the study, the amount of uneaten food was recorded to the nearest quarter; it is estimated that a quarter, half, three-quarters, or the entire portion have been eaten. Macronutrient intake was calculated from the recipes used in the Levit’s Aftercare Centre.

During the second week of this longitudinal crossover study, the diet of patients in both groups was supplemented by 50 g of maltodextrin (Fantomalt, Nutricia, Zoetermeer, The Netherlands) three times a day immediately after the main meal (breakfast at 6:30, lunch at 11:30, dinner at 17:30). Maltodextrin was given to patients diluted in 200 mL of tea, coffee or water, based on patients’ preferences, and the total daily dose of maltodextrin was 150 g. The intake of the drink supplemented with maltodextrin was carefully monitored.

Blood glucose levels were continuously monitored for the entire two-week period by using a subcutaneous sensor Medtronic iPro2 recorder (Medtronic, Prague, Czech Republic). This subcutaneous sensor was used according to the manufacturer’s instructions and all sensors were calibrated four times per day by measuring glycaemia from a capillary blood.

Description data are shown as mean ± standard deviation. The paired Wilcoxon’s test was used for statistical analysis of the comparison of areas under curve (AUCs) of plasma glucose levels during the control week and during the interventional week when maltodextrin was supplemented. Fisher’s exact test was used to compare the frequency of hypo- and hyperglycemias. To analyze the trend over time, simple linear regression was used to assess whether the slope was significantly nonzero.

## 3. Results

In total, 21 out of 22 patients completed the study (14 patients with T2DM and seven patients without diabetes). One patient (nondiabetic) prematurely terminated his participation due to personal reasons. We did not observe any major or even minor adverse side effects of maltodextrin supplementation throughout the study. The administration of oral antidiabetic drugs or the administration of insulin itself was not changed in any patient with T2DM after the addition of maltodextrin to the diet.

The energy and CHO intake were lower, whereas lipid intake was higher in the diabetic group than in the nondiabetic patients. The addition of carbohydrates in the form of maltodextrin was subjectively well tolerated and increased the energy intake in both groups of patients. In the case of diabetic patients, maltodextrin did not significantly alter the intake of macronutrients see Table 1. According to our data, T2DM patients were obese and controls were overweight. We do not have an explanation for this based on our results, but we assume this is perhaps due to the well-known link between obesity and T2DM.

Blood glucose levels were higher in T2DM compared to nondiabetic patients. The addition of maltodextrin immediately after the main meal led to a transient increase in blood glucose levels in both T2DM and nondiabetic patients. The increase in glycaemia after maltodextrin supplementation was more pronounced in patients with diabetes than in nondiabetic patients (see Table 2).

The rise in blood glucose levels after the maltodextrin-supplemented meal was highest in diabetics during the first days of the trial. In the following days, however, the increase in glycaemia after maltodextrin was lower (see Figure 1 and Figure 2).

In addition, the areas under the glycemic curves were higher in T2DM and always increased further after maltodextrin administration in comparison with subjects without diabetes during days 2–4. However, in these patients the significant increase in glycaemia due to added CHO (maltodextrin) was diminished after four days and the effect was absent during days 5–7 (see Figure 3).

Adding maltodextrin to the regular diet gradually reduced and adjusted the blood glucose level. This was apparent especially in T2DM patients. In these patients, the influence of added maltodextrin to the glycemic curve tended to decrease with interval of maltodextrin administration (see Figure 4).

In nondiabetic patients, maltodextrin did not lead to glycemia above 11.1 mmol/L. A blood sugar level above 11.1 mmol/L was more common in diabetics after maltodextrin was administered. However, the increase in blood glucose was transient and only related to the period after maltodextrin administration.

The nocturnal blood glucose level and the area under the curve of blood sugar level did not differ during the period when maltodextrin was added to the diet from the control period without maltodextrin. In nondiabetics, adding maltodextrin led to a more frequent occurrence of low blood glucose levels (below 3.8 mmol/L); however, no patient had clinical signs of hypoglycaemia. In patients with type 2 diabetes, blood glucose levels below 3.8 mmol/L were not observed.

## 4. Discussion

Every disease represents a potential loss of skeletal muscle due to inflammation, inactivity, and reduced nutritional intake. The elderly population, in which physical activity is limited, usually suffers the most from various acute and chronic diseases with subsequent loss of muscle tissue. Early physiotherapy, rehabilitation and most importantly adequate nutrition are essential for the body-wide anabolic response of the organism [3]. During growth and regeneration, our dietary substrates must cover both increased energy expenditure and anabolic processes that are needed to improve physical activity and self-sufficiency. Not only must a sufficient total energy intake be supplied, but specific nutritional substrates are also necessary for tissue regeneration. Carbohydrates are important substrates for the initiation and progression of synthetic and anabolic events with subsequent tissue regeneration. This is related to the fact that glucose and its metabolites are necessary building blocks for other molecules and macromolecules, and participate in various metabolic pathways including NADPH production [9]. Glucose is used for the synthesis of other molecules, which have a fundamental structural and functional importance such as amino acids, nucleic acids and structural molecules such as chondroitin sulphate [15], dermatan sulphate, keratan sulphate, and heparan sulphate; or protein glycosylation [16]. Glycocalyx of the cell membrane [16] is vital for cell physiology, such as adhesion, migration, and cell-to-cell communication, or glycosylated intracellular proteins that are associated with cell membrane translocation and exocytosis [17]. Moreover, the supply of glucose is essential for immune processes and prevention of oxidative stress through the increase of the intracellular pool of NADPH [18]. NADPH is generated in pentose phosphate pathway (PPP), which uses glucose-6-phosphate [19], the main metabolite of glucose. NADPH is used in oxidative killing of bacteria via NADPH oxidase [20,21] and is also important for the regulation of immune responses [20,22]. PPP also generates ribose-5-P, an important precursor for the synthesis of nucleotides or ATP. Precisely for these reasons, glucose is one of the most important nutritional substrates for anabolism [9,10].

The results of our study show that increased carbohydrate intake did not significantly worsen diabetes compensation in elderly patients with type 2 diabetes, even though the administration of oral antidiabetic drugs and/or insulin did not change during the study. Moreover, we found that a higher than presently recommended intake of carbohydrates during the first four days improved some parameters of diabetes compensation during days 5 to 7. The course of glycemia improved during the night when patients were fasting. Blood glucose levels and areas under the glycemic curves were higher in postprandial intervals after combining the main meal with 50 g of maltodextrin compared to the condition when this carbohydrate was not given. However, during the week when maltodextrin was added to the diet, we also noticed a gradual reduction of the mentioned increase in blood sugar after added carbohydrates. This observation appears paradoxical at first, as it could mean that increased carbohydrate intake did not worsen diabetic compensation (no effect on fasting glycaemia) but may even improve carbohydrate tolerance and thus also reduce insulin resistance (gradual reduction of postprandial hyperglycemia). Similar studies in which an improvement of IR was observed after the administration of carbohydrate drinks were published previously. For instance, administration of glucose prior to surgery improved insulin resistance in the postoperative period [23,24]. Some authors have also observed other clinical benefits, such as mental well-being, reduction of postoperative nausea and vomiting, improvement of postoperative muscle metabolism and function, including muscle strength [25,26,27]. However, the exact reason for this metabolic phenomenon is still not completely understood [28].

It must be emphasized that the stores of glucose and glycogen in our body are very low; they are much lower than protein and fat stores. Liver glycogen reserves cover the body’s energy requirements for less than 24 hours [29]. Therefore, the complete glucose oxidation probably occurs only when its supplies and external amount exceeds the need for previously mentioned anabolic metabolic processes [9]. If the intake of Glc is sufficient, part of Glc is redirected to be fully oxidized in the Krebs cycle [10]. If the intake of carbohydrates is insufficient, the complete oxidation of glucose decreases and new glucose formation and turnover increases. In the extreme case of starvation, in which an organism lacks substrate for PPP (i.e., glucose-6-P) all intermediate products are, on the contrary, directed in the opposite direction toward gluconeogenesis [30]. Similar metabolic switch is observed during insulin resistance development. Insulin resistance (IR) is a condition in which more insulin is required to reach a certain plasma glucose concentration than in insulin-sensitive situations. The clinical manifestation of insulin resistance consists in Glc or carbohydrate intolerance in the sense that postprandial hyperglycaemia and hyperinsulinemia occur, followed by fasting hypoglycaemia. IR is associated not only with type 2 diabetes mellitus, but also with severe trauma, infection, sepsis, and other inflammatory conditions. IR occurs also during physiological situations such as rapid growth, puberty, pregnancy, lactation or fasting [31,32]. The common denominator of all these conditions related to insulin resistance is that glucose is redirected to meet needs other than those concerning energy (see above). Surprisingly the role of glucose molecule itself in the mechanism of IR development was still not fully elucidated [33,34].

In our opinion, insulin resistance is a metabolic adaptation that serves to ensure the specific needs of cells and tissues that are active in conditions of stress, inflammation, as well as growth and regeneration; this may indicate an increased, rather than decreased, requirement for glucose [9]. This claim is supported by the results of our study, in which a gradual improvement of postprandial hyperglycaemia was observed during the period when maltodextrin was added to the diet of type 2 diabetics. Data of current study confirm our previous findings where increased glucose intake had beneficial effect on insulin sensitivity in critically ill patients [35].

Limitation of this study is that we did not directly measure the insulin sensitivity by glucose/insulin clamp, but we estimated the insulin resistance from fasting and postprandial glycemic curve. Another limitation is that we did not measure physical parameters that would be interesting to assess, especially in patients undergoing a physiotherapy to increase their physical fitness. On the other hand, during preparation of this study, we did not assume a major effect just after one week of high glucose oral feeding. A big benefit of this study is that we were able to see a longitudinal glycemia at rest and postprandial after each meal in a closely controlled environment on human subjects. Whereas in animal studies performed previously researchers were able to describe more accurately insulin resistance, their conclusions are always based on animal (mostly rodent) metabolism [36]; the natural diet of rodents contains a relatively high proportion of energy in the form of carbohydrates. The benefit of this study is that we had stable patients with no acute disease. In addition, the goal of our patients’ stay was to improve muscle strength and physical activity and, in most patients, to improve the healing of chronic wounds. It is clear that the goal was the induction of an anabolic reaction, which is the basic required effect of enteral nutritional support. Moreover, we were able to measure portions which were eaten by our participants and closely monitor their dietary intake. Our data were not dependent on self-reporting which usually creates a methodological error. We also confirmed our method and used these sensors in a way which was a novelty for authors to the day of performing this study. 

The results of our study show that in T2DM patients who require nutritional support for the purpose of regeneration and improvement of physical activity, the intake of carbohydrates may be more liberal than previously assumed.

## 5. Conclusions

Based on our results, we suggest a more liberal amount of carbohydrates in oral and enteral nutrition products for nutritional support. The amount of carbohydrates in the diet should be increased in enteral nutrition intended for patients in whom the goal of nutritional support is an anabolic reaction and an increase in muscle mass and strength. Further restriction of carbohydrate intake due to insulin resistance tends to contradict the goals of nutritional support described above. The optimal amount of carbohydrates in future products intended for patients with insulin resistance or type 2 diabetes who need nutritional support should thus be the subject of future studies.

## Figures and Tables

**Figure 1 nutrients-15-00439-f001:**
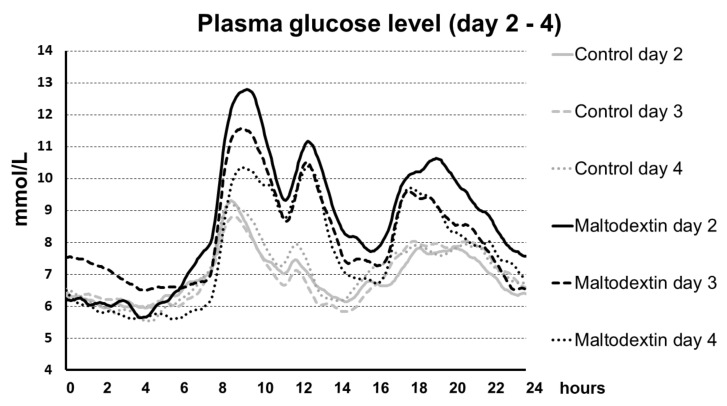
Plasma glucose level during day 2, 3, and 4 in T2DM patients during control period and during maltodextrin administration.

**Figure 2 nutrients-15-00439-f002:**
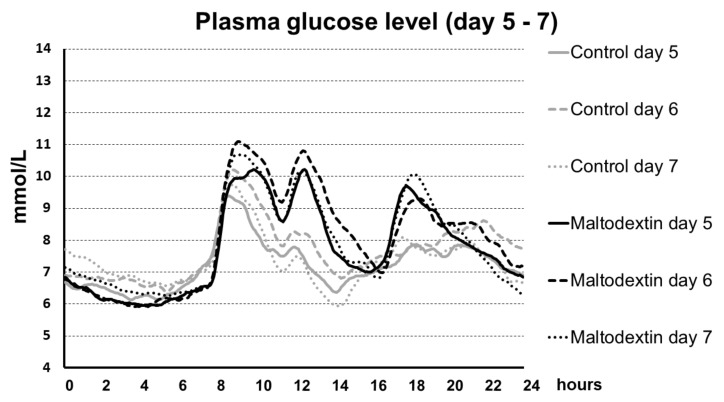
Plasma glucose level during days 5, 6, and 7 in T2DM patients during control period and during maltodextrin administration. It is apparent that the postprandial peak hyperglycemia was blunted in comparison with days 2–4.

**Figure 3 nutrients-15-00439-f003:**
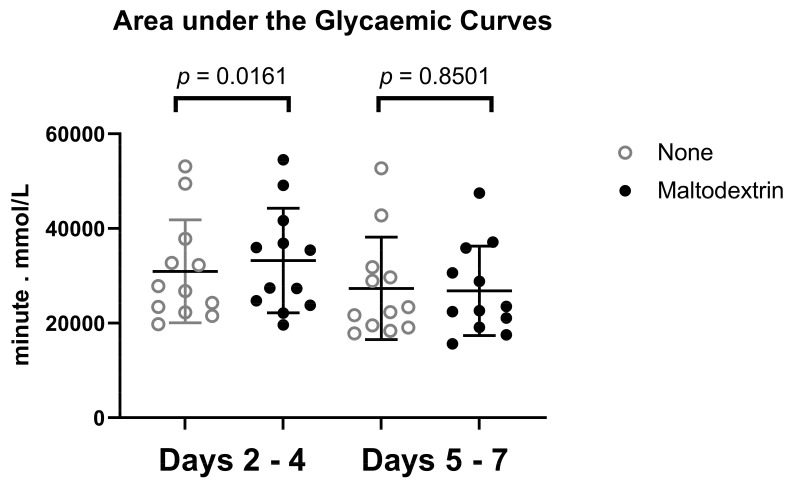
Area under the glycemic curves. A comparison of areas under the glycemic curves: the patients receiving maltodextrin had significantly higher area under the curve during days 2–4 (*p* = 0.0161). This difference was absent during days 5–7 (*p* = 0.8501).

**Figure 4 nutrients-15-00439-f004:**
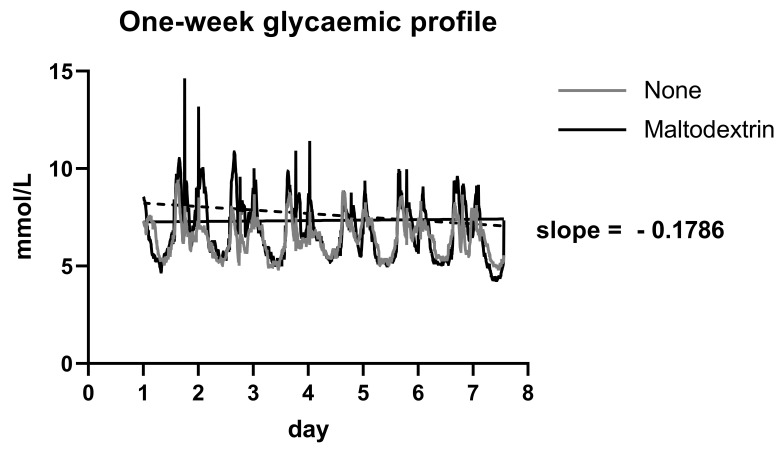
One-week glycemic profile: the patients receiving maltodextrin had higher glycaemia during the postprandial period. This phenomenon was significantly blunted in latter half of the follow-up and the glycaemia in this cohort had a decreasing tendency (slope of −0.1786; *p* < 0.0001).

**Table 1 nutrients-15-00439-t001:** Study groups and intake of energy, CHO, protein and fat in study in T2DM patients and patients without diabetes (* *p* < 0.05, ** *p* < 0.001).

	Type 2 Diabetes	No Diabetes
n (Female)	14 (8)	7 (6)
Age (years)	72.7 ± 9.8	78.9 ± 7.6
Height (cm)	164.6 ± 8.4	163.6 ± 6.0
Weight (kg)	85.8 ± 20.3	72.6 ± 6.2 *
	**Standard diet**	
Energy (kcal/day)	2082.5 ± 114.0	2219.1 ± 117.7 *
Carbohydrate (g/day)	233.4 ± 20.8	317.1 ± 24.3 **
Protein (g/day)	90.0 ± 12.6	84.0 ± 3.9
Fat (g/day)	87.6 ± 3.6	68.3 ± 6.5 **
	**Standard diet + maltodextrin**	
Energy (kcal/day)	2642.7 ± 120.2	2839.6 ± 133.5 *
Carbohydrate (g/day)	384.0 ± 16.6	467.1 ± 24.3 **
Protein (g/day)	88.6 ± 9.4	85.6 ± 3.8
Fat (g/day)	83.6 ± 6.4	69.9 ± 4.2 **

**Table 2 nutrients-15-00439-t002:** Effects of maltodextrin addition on mean glycaemia in diabetic and nondiabetic patients.

T2DM Patients		Plasma Glucose Levels (mmol/L)		
	**Time interval**	**Control**	**Maltodextrin**	** *p* **
	0:00–6:30	6.37 ± 2.54	6.21 ± 2.17	0.206
	6:30–10:00	8.19 ± 2.47	8.93 ± 2.90	0.005
	10:00–11:30	7.66 ± 2.83	9.71 ± 4.21	<0.001
	11:30–14:30	6.90 ± 2.02	9.34 ± 3.50	<0.001
	14:30–17:30	6.99 ± 1.81	7.67 ± 2.42	0.034
	17:30–20:30	7.77 ± 2.29	9.16 ± 2.85	<0.001
	20:30–0:00	7.40 ± 2.93	7.60 ± 3.09	0.313
**Nondiabetic patients**				
	**Time interval**	**Control**	**Maltodextrin**	** *p* **
	0:00–6:30	5.22 ± 0.30	4.83 ± 0.31	0.01
	6:30–10:00	6.06 ± 0.39	5.97 ± 0.55	0.263
	10:00–11:30	5.83 ± 0.38	5.80 ± 0.69	0.454
	11:30–14:30	5.94 ± 0.43	6.10 ± 0.61	0.177
	14:30–17:30	6.13 ± 0.38	5.94 ± 0.30	0.128
	17:30–20:30	5.96 ± 0.38	6.10 ± 0.34	0.158
	20:30–0:00	5.54 ± 0.23	5.34 ± 0.27	0.140

## Data Availability

The data supporting reported results can be obtained from corresponding author Lubos Sobotka.

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
