# Peer review of "Should Carbohydrate Intake Be More Liberal during Oral and Enteral Nutrition in Type 2 Diabetic Patients?"

_nutrients, 2023, doi:10.3390/nu15020439_

Round 1
Reviewer 1 Report
General comments
This original article deals with an important and innovative topic. The authors investigate to what extent an increased intake of a rapidly digestible oligosaccharide/polysaccharide maltodextrin affect blood glucose levels. Such Studies determining the effect of increased carbohydrate intake on the compensation of type 2 diabetes may serve for better understanding of pathophysiologic aspects and for future development of oral/enteral nutrition products optimizing the nutritional support in malnourished diabetic or insulin-resistant patients.
There are several points that need to be addressed in this manuscript:
- The authors also write enteral nutrition in the title, but the study only deals with oral nutrition. I would adapt the title and write oral nutrition instead of enteral nutrition in the text everywhere when appropriate. That way, readers will be less confused.
- Malnutrition is mentioned in the first sentence of the abstract and in the last sentence of the conclusions. However, throughout the manuscript there is no discussion of malnutrition or definition of malnutrition, nor is there any description of whether the patients were malnourished or not.
- Both populations, with or without diabetes mellitus, are not sufficiently characterized. Important factors such as medication, especially diabetes medication and comorbidities are completely missing. These would need listed in a table and also briefly discussed in the discussion, as these factors are very relevant to the results of this study.
- Where the patients are hospitalised is unclear for someone who does not know the Levit's Aftercare Centre in Horice. Are they acutely hospitalised patients or chronic patients in a rehabilitation clinic or in a nursing home or in an elderly home? - The authors mentioned that the daily energy delivered by the usual hospital meals are approximately 2’000 kcal consisting in 250-270 g of CHOs, 75-100 g of proteins and 65-77 g of lipids. The carbohydrate intake seems a little high to me, but I think the authors are following evidence-based recommendations and can explain this.
Specific comments
Abstract
- abstract so far no remarks
Introduction
- - Line 54-57: Authors should provide a reference of these recommendations.
Methods
- - Study design: Are you sure that the design is crossover? What exactly is crossing?
- - Inclusion criteria are missing
- - Did the patients sign an informed consent?
- - Exclusion criteria are missing
- - Lines 89-90: You mentioned non-diabetic patients get 350 g CHO, in the same sentence you write that non-diabetic patients received 250 g CHO. This is confusing. What is correct? Please clarify.
- What was the reason for the hospitalization of the patients?
- Were the patients malnourished? How did you assess malnutrition and sarcopenia?
- Please, give a more detailed description of the studied population: Comorbidities? Medication? Antidiabetic medication? Insulin therapy?
Results
- Please add a table with malnutrition state, sarcopenia, comorbidities and medication of the patients The result section should be elaborated more in detail.
- Please mention/discuss why the diabetic patients were obese and the controls merely overweight?
Discussion
- The first 2 paragraphs (Lines 167-203) in the discussion do not add any further value to the study, so I would delete them.
- Please revise the rest of the discussion and focus on the results.
- Strengths and limitations of the study are missing. Please add accordingly
Conclusion
- Conclusions so far in order.
Author Response
The answers to the reviewer 1:
General comments
This original article deals with an important and innovative topic. The authors investigate to what extent an increased intake of a rapidly digestible oligosaccharide/polysaccharide maltodextrin affect blood glucose levels. Such Studies determining the effect of increased carbohydrate intake on the compensation of type 2 diabetes may serve for better understanding of pathophysiologic aspects and for future development of oral/enteral nutrition products optimizing the nutritional support in malnourished diabetic or insulin-resistant patients.
The points that are addressed in this manuscript:
The authors also write enteral nutrition in the title, but the study only deals with oral nutrition. I would adapt the title and write oral nutrition instead of enteral nutrition in the text everywhere when appropriate. That way, readers will be less confused.
Thank you for this comment. We agree with reviewer, but due to the special issue in which is this article devoted to (focused on enteral nutrition) we want to emphasize the link that not only oral feeding, but also enteral nutrition is relevant for our results. We have changed text appropriately and add “oral and enteral nutrition” in respective places in the heading and in text.
Malnutrition is mentioned in the first sentence of the abstract and in the last sentence of the conclusions. However, throughout the manuscript there is no discussion of malnutrition or definition of malnutrition, nor is there any description of whether the patients were malnourished or not.
We agree with reviewer, there is no discussion of malnutrition or its definition and since we did not estimate malnutrition by our methods, we decided to remove malnutrition in mentioned places in text and make minor changes in text instead.
Both populations, with or without diabetes mellitus, are not sufficiently characterized. Important factors such as medication, especially diabetes medication and comorbidities are completely missing. These would need listed in a table and also briefly discussed in the discussion, as these factors are very relevant to the results of this study.
Thank the reviewer for this comment. We decided to include a supplementary table 1 in which we show requested information (see supplementary table 1)
Where the patients are hospitalised is unclear for someone who does not know the Levit’s Aftercare Centre in Horice. Are they acutely hospitalised patients or chronic patients in a rehabilitation clinic or in a nursing home or in an elderly home? - The authors mentioned that the daily energy delivered by the usual hospital meals are approximately 2’000 kcal consisting in 250-270 g of CHOs, 75-100 g of proteins and 65-77 g of lipids. The carbohydrate intake seems a little high to me, but I think the authors are following evidence-based recommendations and can explain this.
The Levit centre is now more characterised in the text. The composition of macronutrient in the diet both for diabetic and non-diabetic patients was in accordance with hospital dietary system in rehabilitation centres in Czech Republic (see ref 35).
Specific comments
Abstract
- Abstract so far no remarks
Introduction
- Line 54-57: Authors should provide a reference of these recommendations.
We added a relevant reference in text (see ref 35.)
Methods
- Study design: Are you sure that the design is crossover, what exactly is crossing
Crossover study is a longitudinal study in which subjects receive a sequence of different treatments. In our study each patient received both control and maltodextrin enriched diets in first and second week respectively. Please see changes in text below.
- Inclusion criteria are missing
- Did the patients sign an informed consent?
- Exclusion criteria are missing
Thank you for pointing out the lack of this important information, we made changes in method section (please see text below).
- Lines 89-90: You mentioned non-diabetic patients get 350 g CHO, in the same sentence you write that non-diabetic patients received 250 g CHO. This is confusing. What is correct? Please clarify.
Thank you for your comment that is our mistake. The correct sentence is: Maximum daily carbohydrate intake in the standard diet for non-diabetics was 350 g (57 % of total energy intake) and 250 g (44 % of total energy intake) for T2DM patients. We corrected this sentence in the text.
- What was the reason for the hospitalization of the patients?
Patients were hospitalized in order to improve their general condition after an acute illness or to heal chronic wounds. Physical rehabilitation and nutritional care were the main methods of treatment. Patients with wounds underwent local treatment of chronic wounds. We added this information to the text and in the supplementary table (see Supp. table 1).
- Were the patients malnourished? How did you assess malnutrition and sarcopenia?
No, malnourished patients requiring nutritional support were excluded. The patients were regularly screened with NRS 2002 screening tool.
- Please, give a more detailed description of the studied population: Comorbidities? Medication? Antidiabetic medication? Insulin therapy?
Please see the supplementary table.
Results
- Please add a table with malnutrition state, sarcopenia, comorbidities and medication of the patients
- The result section should be elaborated more in detail
Thank you for this comment, we added a supplementary table as mentioned above with requested information.
- Please mention/discuss why the diabetic patients were obese and the controls merely overweight?
Thank you for this comment, but we cannot explain the reason for this. The patients with diabetes were overweight perhaps due to a known link between obesity and T2DM. We added a sentence to the text.
Discussion
- The first 2 paragraphs (Lines 167-203) in the discussion do not add any further value to the study, so I would delete them.
- Please revise the rest of the discussion and focus on the results.
- Strengths and limitations of the study are missing. Please add accordingly
Thank you for these important comments. We added limitations, benefits and novelty aspect of our study into the text. We agree with the reviewer, that first two paragraphs are unnecessarily long, but we consider it important to support our conclusions. We edited requested paragraphs and made them shorter.
Conclusion
Conclusions so far in order.
The answers to the reviewer 1:
General comments
This original article deals with an important and innovative topic. The authors investigate to what extent an increased intake of a rapidly digestible oligosaccharide/polysaccharide maltodextrin affect blood glucose levels. Such Studies determining the effect of increased carbohydrate intake on the compensation of type 2 diabetes may serve for better understanding of pathophysiologic aspects and for future development of oral/enteral nutrition products optimizing the nutritional support in malnourished diabetic or insulin-resistant patients.
The points that are addressed in this manuscript:
The authors also write enteral nutrition in the title, but the study only deals with oral nutrition. I would adapt the title and write oral nutrition instead of enteral nutrition in the text everywhere when appropriate. That way, readers will be less confused.
Thank you for this comment. We agree with reviewer, but due to the special issue in which is this article devoted to (focused on enteral nutrition) we want to emphasize the link that not only oral feeding, but also enteral nutrition is relevant for our results. We have changed text appropriately and add “oral and enteral nutrition” in respective places in the heading and in text.
Malnutrition is mentioned in the first sentence of the abstract and in the last sentence of the conclusions. However, throughout the manuscript there is no discussion of malnutrition or definition of malnutrition, nor is there any description of whether the patients were malnourished or not.
We agree with reviewer, there is no discussion of malnutrition or its definition and since we did not estimate malnutrition by our methods, we decided to remove malnutrition in mentioned places in text and make minor changes in text instead.
Both populations, with or without diabetes mellitus, are not sufficiently characterized. Important factors such as medication, especially diabetes medication and comorbidities are completely missing. These would need listed in a table and also briefly discussed in the discussion, as these factors are very relevant to the results of this study.
Thank the reviewer for this comment. We decided to include a supplementary table 1 in which we show requested information (see supplementary table 1)
Where the patients are hospitalised is unclear for someone who does not know the Levit’s Aftercare Centre in Horice. Are they acutely hospitalised patients or chronic patients in a rehabilitation clinic or in a nursing home or in an elderly home? - The authors mentioned that the daily energy delivered by the usual hospital meals are approximately 2’000 kcal consisting in 250-270 g of CHOs, 75-100 g of proteins and 65-77 g of lipids. The carbohydrate intake seems a little high to me, but I think the authors are following evidence-based recommendations and can explain this.
The Levit centre is now more characterised in the text. The composition of macronutrient in the diet both for diabetic and non-diabetic patients was in accordance with hospital dietary system in rehabilitation centres in Czech Republic (see ref 35).
Specific comments
Abstract
- Abstract so far no remarks
Introduction
- Line 54-57: Authors should provide a reference of these recommendations.
We added a relevant reference in text (see ref 35.)
Methods
- Study design: Are you sure that the design is crossover, what exactly is crossing
Crossover study is a longitudinal study in which subjects receive a sequence of different treatments. In our study each patient received both control and maltodextrin enriched diets in first and second week respectively. Please see changes in text below.
- Inclusion criteria are missing
- Did the patients sign an informed consent?
- Exclusion criteria are missing
Thank you for pointing out the lack of this important information, we made changes in method section (please see text below).
- Lines 89-90: You mentioned non-diabetic patients get 350 g CHO, in the same sentence you write that non-diabetic patients received 250 g CHO. This is confusing. What is correct? Please clarify.
Thank you for your comment that is our mistake. The correct sentence is: Maximum daily carbohydrate intake in the standard diet for non-diabetics was 350 g (57 % of total energy intake) and 250 g (44 % of total energy intake) for T2DM patients. We corrected this sentence in the text.
- What was the reason for the hospitalization of the patients?
Patients were hospitalized in order to improve their general condition after an acute illness or to heal chronic wounds. Physical rehabilitation and nutritional care were the main methods of treatment. Patients with wounds underwent local treatment of chronic wounds. We added this information to the text and in the supplementary table (see Supp. table 1).
- Were the patients malnourished? How did you assess malnutrition and sarcopenia?
No, malnourished patients requiring nutritional support were excluded. The patients were regularly screened with NRS 2002 screening tool.
- Please, give a more detailed description of the studied population: Comorbidities? Medication? Antidiabetic medication? Insulin therapy?
Please see the supplementary table.
Results
- Please add a table with malnutrition state, sarcopenia, comorbidities and medication of the patients
- The result section should be elaborated more in detail
Thank you for this comment, we added a supplementary table as mentioned above with requested information.
- Please mention/discuss why the diabetic patients were obese and the controls merely overweight?
Thank you for this comment, but we cannot explain the reason for this. The patients with diabetes were overweight perhaps due to a known link between obesity and T2DM. We added a sentence to the text.
Discussion
- The first 2 paragraphs (Lines 167-203) in the discussion do not add any further value to the study, so I would delete them.
- Please revise the rest of the discussion and focus on the results.
- Strengths and limitations of the study are missing. Please add accordingly
Thank you for these important comments. We added limitations, benefits and novelty aspect of our study into the text. We agree with the reviewer, that first two paragraphs are unnecessarily long, but we consider it important to support our conclusions. We edited requested paragraphs and made them shorter.
Conclusion
Conclusions so far in order.
The answers to the reviewer 1:
General comments
This original article deals with an important and innovative topic. The authors investigate to what extent an increased intake of a rapidly digestible oligosaccharide/polysaccharide maltodextrin affect blood glucose levels. Such Studies determining the effect of increased carbohydrate intake on the compensation of type 2 diabetes may serve for better understanding of pathophysiologic aspects and for future development of oral/enteral nutrition products optimizing the nutritional support in malnourished diabetic or insulin-resistant patients.
The points that are addressed in this manuscript:
The authors also write enteral nutrition in the title, but the study only deals with oral nutrition. I would adapt the title and write oral nutrition instead of enteral nutrition in the text everywhere when appropriate. That way, readers will be less confused.
Thank you for this comment. We agree with reviewer, but due to the special issue in which is this article devoted to (focused on enteral nutrition) we want to emphasize the link that not only oral feeding, but also enteral nutrition is relevant for our results. We have changed text appropriately and add “oral and enteral nutrition” in respective places in the heading and in text.
Malnutrition is mentioned in the first sentence of the abstract and in the last sentence of the conclusions. However, throughout the manuscript there is no discussion of malnutrition or definition of malnutrition, nor is there any description of whether the patients were malnourished or not.
We agree with reviewer, there is no discussion of malnutrition or its definition and since we did not estimate malnutrition by our methods, we decided to remove malnutrition in mentioned places in text and make minor changes in text instead.
Both populations, with or without diabetes mellitus, are not sufficiently characterized. Important factors such as medication, especially diabetes medication and comorbidities are completely missing. These would need listed in a table and also briefly discussed in the discussion, as these factors are very relevant to the results of this study.
Thank the reviewer for this comment. We decided to include a supplementary table 1 in which we show requested information (see supplementary table 1)
Where the patients are hospitalised is unclear for someone who does not know the Levit’s Aftercare Centre in Horice. Are they acutely hospitalised patients or chronic patients in a rehabilitation clinic or in a nursing home or in an elderly home? - The authors mentioned that the daily energy delivered by the usual hospital meals are approximately 2’000 kcal consisting in 250-270 g of CHOs, 75-100 g of proteins and 65-77 g of lipids. The carbohydrate intake seems a little high to me, but I think the authors are following evidence-based recommendations and can explain this.
The Levit centre is now more characterised in the text. The composition of macronutrient in the diet both for diabetic and non-diabetic patients was in accordance with hospital dietary system in rehabilitation centres in Czech Republic (see ref 35).
Specific comments
Abstract
- Abstract so far no remarks
Introduction
- Line 54-57: Authors should provide a reference of these recommendations.
We added a relevant reference in text (see ref 35.)
Methods
- Study design: Are you sure that the design is crossover, what exactly is crossing
Crossover study is a longitudinal study in which subjects receive a sequence of different treatments. In our study each patient received both control and maltodextrin enriched diets in first and second week respectively. Please see changes in text below.
- Inclusion criteria are missing
- Did the patients sign an informed consent?
- Exclusion criteria are missing
Thank you for pointing out the lack of this important information, we made changes in method section (please see text below).
- Lines 89-90: You mentioned non-diabetic patients get 350 g CHO, in the same sentence you write that non-diabetic patients received 250 g CHO. This is confusing. What is correct? Please clarify.
Thank you for your comment that is our mistake. The correct sentence is: Maximum daily carbohydrate intake in the standard diet for non-diabetics was 350 g (57 % of total energy intake) and 250 g (44 % of total energy intake) for T2DM patients. We corrected this sentence in the text.
- What was the reason for the hospitalization of the patients?
Patients were hospitalized in order to improve their general condition after an acute illness or to heal chronic wounds. Physical rehabilitation and nutritional care were the main methods of treatment. Patients with wounds underwent local treatment of chronic wounds. We added this information to the text and in the supplementary table (see Supp. table 1).
- Were the patients malnourished? How did you assess malnutrition and sarcopenia?
No, malnourished patients requiring nutritional support were excluded. The patients were regularly screened with NRS 2002 screening tool.
- Please, give a more detailed description of the studied population: Comorbidities? Medication? Antidiabetic medication? Insulin therapy?
Please see the supplementary table.
Results
- Please add a table with malnutrition state, sarcopenia, comorbidities and medication of the patients
- The result section should be elaborated more in detail
Thank you for this comment, we added a supplementary table as mentioned above with requested information.
- Please mention/discuss why the diabetic patients were obese and the controls merely overweight?
Thank you for this comment, but we cannot explain the reason for this. The patients with diabetes were overweight perhaps due to a known link between obesity and T2DM. We added a sentence to the text.
Discussion
- The first 2 paragraphs (Lines 167-203) in the discussion do not add any further value to the study, so I would delete them.
- Please revise the rest of the discussion and focus on the results.
- Strengths and limitations of the study are missing. Please add accordingly
Thank you for these important comments. We added limitations, benefits and novelty aspect of our study into the text. We agree with the reviewer, that first two paragraphs are unnecessarily long, but we consider it important to support our conclusions. We edited requested paragraphs and made them shorter.
Conclusion
Conclusions so far in order.
The answers to the reviewer 1:
General comments
This original article deals with an important and innovative topic. The authors investigate to what extent an increased intake of a rapidly digestible oligosaccharide/polysaccharide maltodextrin affect blood glucose levels. Such Studies determining the effect of increased carbohydrate intake on the compensation of type 2 diabetes may serve for better understanding of pathophysiologic aspects and for future development of oral/enteral nutrition products optimizing the nutritional support in malnourished diabetic or insulin-resistant patients.
The points that are addressed in this manuscript:
The authors also write enteral nutrition in the title, but the study only deals with oral nutrition. I would adapt the title and write oral nutrition instead of enteral nutrition in the text everywhere when appropriate. That way, readers will be less confused.
Thank you for this comment. We agree with reviewer, but due to the special issue in which is this article devoted to (focused on enteral nutrition) we want to emphasize the link that not only oral feeding, but also enteral nutrition is relevant for our results. We have changed text appropriately and add “oral and enteral nutrition” in respective places in the heading and in text.
Malnutrition is mentioned in the first sentence of the abstract and in the last sentence of the conclusions. However, throughout the manuscript there is no discussion of malnutrition or definition of malnutrition, nor is there any description of whether the patients were malnourished or not.
We agree with reviewer, there is no discussion of malnutrition or its definition and since we did not estimate malnutrition by our methods, we decided to remove malnutrition in mentioned places in text and make minor changes in text instead.
Both populations, with or without diabetes mellitus, are not sufficiently characterized. Important factors such as medication, especially diabetes medication and comorbidities are completely missing. These would need listed in a table and also briefly discussed in the discussion, as these factors are very relevant to the results of this study.
Thank the reviewer for this comment. We decided to include a supplementary table 1 in which we show requested information (see supplementary table 1)
Where the patients are hospitalised is unclear for someone who does not know the Levit’s Aftercare Centre in Horice. Are they acutely hospitalised patients or chronic patients in a rehabilitation clinic or in a nursing home or in an elderly home? - The authors mentioned that the daily energy delivered by the usual hospital meals are approximately 2’000 kcal consisting in 250-270 g of CHOs, 75-100 g of proteins and 65-77 g of lipids. The carbohydrate intake seems a little high to me, but I think the authors are following evidence-based recommendations and can explain this.
The Levit centre is now more characterised in the text. The composition of macronutrient in the diet both for diabetic and non-diabetic patients was in accordance with hospital dietary system in rehabilitation centres in Czech Republic (see ref 35).
Specific comments
Abstract
- Abstract so far no remarks
Introduction
- Line 54-57: Authors should provide a reference of these recommendations.
We added a relevant reference in text (see ref 35.)
Methods
- Study design: Are you sure that the design is crossover, what exactly is crossing
Crossover study is a longitudinal study in which subjects receive a sequence of different treatments. In our study each patient received both control and maltodextrin enriched diets in first and second week respectively. Please see changes in text below.
- Inclusion criteria are missing
- Did the patients sign an informed consent?
- Exclusion criteria are missing
Thank you for pointing out the lack of this important information, we made changes in method section (please see text below).
- Lines 89-90: You mentioned non-diabetic patients get 350 g CHO, in the same sentence you write that non-diabetic patients received 250 g CHO. This is confusing. What is correct? Please clarify.
Thank you for your comment that is our mistake. The correct sentence is: Maximum daily carbohydrate intake in the standard diet for non-diabetics was 350 g (57 % of total energy intake) and 250 g (44 % of total energy intake) for T2DM patients. We corrected this sentence in the text.
- What was the reason for the hospitalization of the patients?
Patients were hospitalized in order to improve their general condition after an acute illness or to heal chronic wounds. Physical rehabilitation and nutritional care were the main methods of treatment. Patients with wounds underwent local treatment of chronic wounds. We added this information to the text and in the supplementary table (see Supp. table 1).
- Were the patients malnourished? How did you assess malnutrition and sarcopenia?
No, malnourished patients requiring nutritional support were excluded. The patients were regularly screened with NRS 2002 screening tool.
- Please, give a more detailed description of the studied population: Comorbidities? Medication? Antidiabetic medication? Insulin therapy?
Please see the supplementary table.
Results
- Please add a table with malnutrition state, sarcopenia, comorbidities and medication of the patients
- The result section should be elaborated more in detail
Thank you for this comment, we added a supplementary table as mentioned above with requested information.
- Please mention/discuss why the diabetic patients were obese and the controls merely overweight?
Thank you for this comment, but we cannot explain the reason for this. The patients with diabetes were overweight perhaps due to a known link between obesity and T2DM. We added a sentence to the text.
Discussion
- The first 2 paragraphs (Lines 167-203) in the discussion do not add any further value to the study, so I would delete them.
- Please revise the rest of the discussion and focus on the results.
- Strengths and limitations of the study are missing. Please add accordingly
Thank you for these important comments. We added limitations, benefits and novelty aspect of our study into the text. We agree with the reviewer, that first two paragraphs are unnecessarily long, but we consider it important to support our conclusions. We edited requested paragraphs and made them shorter.
Conclusion
Conclusions so far in order.
The answers to the reviewer 1:
General comments
This original article deals with an important and innovative topic. The authors investigate to what extent an increased intake of a rapidly digestible oligosaccharide/polysaccharide maltodextrin affect blood glucose levels. Such Studies determining the effect of increased carbohydrate intake on the compensation of type 2 diabetes may serve for better understanding of pathophysiologic aspects and for future development of oral/enteral nutrition products optimizing the nutritional support in malnourished diabetic or insulin-resistant patients.
The points that are addressed in this manuscript:
The authors also write enteral nutrition in the title, but the study only deals with oral nutrition. I would adapt the title and write oral nutrition instead of enteral nutrition in the text everywhere when appropriate. That way, readers will be less confused.
Thank you for this comment. We agree with reviewer, but due to the special issue in which is this article devoted to (focused on enteral nutrition) we want to emphasize the link that not only oral feeding, but also enteral nutrition is relevant for our results. We have changed text appropriately and add “oral and enteral nutrition” in respective places in the heading and in text.
Malnutrition is mentioned in the first sentence of the abstract and in the last sentence of the conclusions. However, throughout the manuscript there is no discussion of malnutrition or definition of malnutrition, nor is there any description of whether the patients were malnourished or not.
We agree with reviewer, there is no discussion of malnutrition or its definition and since we did not estimate malnutrition by our methods, we decided to remove malnutrition in mentioned places in text and make minor changes in text instead.
Both populations, with or without diabetes mellitus, are not sufficiently characterized. Important factors such as medication, especially diabetes medication and comorbidities are completely missing. These would need listed in a table and also briefly discussed in the discussion, as these factors are very relevant to the results of this study.
Thank the reviewer for this comment. We decided to include a supplementary table 1 in which we show requested information (see supplementary table 1)
Where the patients are hospitalised is unclear for someone who does not know the Levit’s Aftercare Centre in Horice. Are they acutely hospitalised patients or chronic patients in a rehabilitation clinic or in a nursing home or in an elderly home? - The authors mentioned that the daily energy delivered by the usual hospital meals are approximately 2’000 kcal consisting in 250-270 g of CHOs, 75-100 g of proteins and 65-77 g of lipids. The carbohydrate intake seems a little high to me, but I think the authors are following evidence-based recommendations and can explain this.
The Levit centre is now more characterised in the text. The composition of macronutrient in the diet both for diabetic and non-diabetic patients was in accordance with hospital dietary system in rehabilitation centres in Czech Republic (see ref 35).
Specific comments
Abstract
- Abstract so far no remarks
Introduction
- Line 54-57: Authors should provide a reference of these recommendations.
We added a relevant reference in text (see ref 35.)
Methods
- Study design: Are you sure that the design is crossover, what exactly is crossing
Crossover study is a longitudinal study in which subjects receive a sequence of different treatments. In our study each patient received both control and maltodextrin enriched diets in first and second week respectively. Please see changes in text below.
- Inclusion criteria are missing
- Did the patients sign an informed consent?
- Exclusion criteria are missing
Thank you for pointing out the lack of this important information, we made changes in method section (please see text below).
- Lines 89-90: You mentioned non-diabetic patients get 350 g CHO, in the same sentence you write that non-diabetic patients received 250 g CHO. This is confusing. What is correct? Please clarify.
Thank you for your comment that is our mistake. The correct sentence is: Maximum daily carbohydrate intake in the standard diet for non-diabetics was 350 g (57 % of total energy intake) and 250 g (44 % of total energy intake) for T2DM patients. We corrected this sentence in the text.
- What was the reason for the hospitalization of the patients?
Patients were hospitalized in order to improve their general condition after an acute illness or to heal chronic wounds. Physical rehabilitation and nutritional care were the main methods of treatment. Patients with wounds underwent local treatment of chronic wounds. We added this information to the text and in the supplementary table (see Supp. table 1).
- Were the patients malnourished? How did you assess malnutrition and sarcopenia?
No, malnourished patients requiring nutritional support were excluded. The patients were regularly screened with NRS 2002 screening tool.
- Please, give a more detailed description of the studied population: Comorbidities? Medication? Antidiabetic medication? Insulin therapy?
Please see the supplementary table.
Results
- Please add a table with malnutrition state, sarcopenia, comorbidities and medication of the patients
- The result section should be elaborated more in detail
Thank you for this comment, we added a supplementary table as mentioned above with requested information.
- Please mention/discuss why the diabetic patients were obese and the controls merely overweight?
Thank you for this comment, but we cannot explain the reason for this. The patients with diabetes were overweight perhaps due to a known link between obesity and T2DM. We added a sentence to the text.
Discussion
- The first 2 paragraphs (Lines 167-203) in the discussion do not add any further value to the study, so I would delete them.
- Please revise the rest of the discussion and focus on the results.
- Strengths and limitations of the study are missing. Please add accordingly
Thank you for these important comments. We added limitations, benefits and novelty aspect of our study into the text. We agree with the reviewer, that first two paragraphs are unnecessarily long, but we consider it important to support our conclusions. We edited requested paragraphs and made them shorter.
Conclusion
Conclusions so far in order.
The answers to the reviewer 1:
General comments
This original article deals with an important and innovative topic. The authors investigate to what extent an increased intake of a rapidly digestible oligosaccharide/polysaccharide maltodextrin affect blood glucose levels. Such Studies determining the effect of increased carbohydrate intake on the compensation of type 2 diabetes may serve for better understanding of pathophysiologic aspects and for future development of oral/enteral nutrition products optimizing the nutritional support in malnourished diabetic or insulin-resistant patients.
The points that are addressed in this manuscript:
The authors also write enteral nutrition in the title, but the study only deals with oral nutrition. I would adapt the title and write oral nutrition instead of enteral nutrition in the text everywhere when appropriate. That way, readers will be less confused.
Thank you for this comment. We agree with reviewer, but due to the special issue in which is this article devoted to (focused on enteral nutrition) we want to emphasize the link that not only oral feeding, but also enteral nutrition is relevant for our results. We have changed text appropriately and add “oral and enteral nutrition” in respective places in the heading and in text.
Malnutrition is mentioned in the first sentence of the abstract and in the last sentence of the conclusions. However, throughout the manuscript there is no discussion of malnutrition or definition of malnutrition, nor is there any description of whether the patients were malnourished or not.
We agree with reviewer, there is no discussion of malnutrition or its definition and since we did not estimate malnutrition by our methods, we decided to remove malnutrition in mentioned places in text and make minor changes in text instead.
Both populations, with or without diabetes mellitus, are not sufficiently characterized. Important factors such as medication, especially diabetes medication and comorbidities are completely missing. These would need listed in a table and also briefly discussed in the discussion, as these factors are very relevant to the results of this study.
Thank the reviewer for this comment. We decided to include a supplementary table 1 in which we show requested information (see supplementary table 1)
Where the patients are hospitalised is unclear for someone who does not know the Levit’s Aftercare Centre in Horice. Are they acutely hospitalised patients or chronic patients in a rehabilitation clinic or in a nursing home or in an elderly home? - The authors mentioned that the daily energy delivered by the usual hospital meals are approximately 2’000 kcal consisting in 250-270 g of CHOs, 75-100 g of proteins and 65-77 g of lipids. The carbohydrate intake seems a little high to me, but I think the authors are following evidence-based recommendations and can explain this.
The Levit centre is now more characterised in the text. The composition of macronutrient in the diet both for diabetic and non-diabetic patients was in accordance with hospital dietary system in rehabilitation centres in Czech Republic (see ref 35).
Specific comments
Abstract
- Abstract so far no remarks
Introduction
- Line 54-57: Authors should provide a reference of these recommendations.
We added a relevant reference in text (see ref 35.)
Methods
- Study design: Are you sure that the design is crossover, what exactly is crossing
Crossover study is a longitudinal study in which subjects receive a sequence of different treatments. In our study each patient received both control and maltodextrin enriched diets in first and second week respectively. Please see changes in text below.
- Inclusion criteria are missing
- Did the patients sign an informed consent?
- Exclusion criteria are missing
Thank you for pointing out the lack of this important information, we made changes in method section (please see text below).
- Lines 89-90: You mentioned non-diabetic patients get 350 g CHO, in the same sentence you write that non-diabetic patients received 250 g CHO. This is confusing. What is correct? Please clarify.
Thank you for your comment that is our mistake. The correct sentence is: Maximum daily carbohydrate intake in the standard diet for non-diabetics was 350 g (57 % of total energy intake) and 250 g (44 % of total energy intake) for T2DM patients. We corrected this sentence in the text.
- What was the reason for the hospitalization of the patients?
Patients were hospitalized in order to improve their general condition after an acute illness or to heal chronic wounds. Physical rehabilitation and nutritional care were the main methods of treatment. Patients with wounds underwent local treatment of chronic wounds. We added this information to the text and in the supplementary table (see Supp. table 1).
- Were the patients malnourished? How did you assess malnutrition and sarcopenia?
No, malnourished patients requiring nutritional support were excluded. The patients were regularly screened with NRS 2002 screening tool.
- Please, give a more detailed description of the studied population: Comorbidities? Medication? Antidiabetic medication? Insulin therapy?
Please see the supplementary table.
Results
- Please add a table with malnutrition state, sarcopenia, comorbidities and medication of the patients
- The result section should be elaborated more in detail
Thank you for this comment, we added a supplementary table as mentioned above with requested information.
- Please mention/discuss why the diabetic patients were obese and the controls merely overweight?
Thank you for this comment, but we cannot explain the reason for this. The patients with diabetes were overweight perhaps due to a known link between obesity and T2DM. We added a sentence to the text.
Discussion
- The first 2 paragraphs (Lines 167-203) in the discussion do not add any further value to the study, so I would delete them.
- Please revise the rest of the discussion and focus on the results.
- Strengths and limitations of the study are missing. Please add accordingly
Thank you for these important comments. We added limitations, benefits and novelty aspect of our study into the text. We agree with the reviewer, that first two paragraphs are unnecessarily long, but we consider it important to support our conclusions. We edited requested paragraphs and made them shorter.
Conclusion
Conclusions so far in order.
Reviewer 2 Report
Please see the attachment.

Author Response
The answers to the reviewer 2:
Title: Should carbohydrate intake be liberal during enteral nutrition in type 2 diabetic patients? Comments to Authors Although the study looks interesting, there are issues with the following findings:
Introduction
- The authors should clarify the novelty of this article in the ‘Introduction’ and ‘Conclusion’ section.
We added a sentence to the text to specify strengths and novelty of this work in the end of discussion section. Please see below changes in text.
- Is there any evidence of in vitro or animal model studies? If this is the case, you must discuss it in the introduction and discussion sections. If not, how did you get involved in human research?
Yes, it was documented using in vitro and animal studies previously. Well summarized in review Campbell 2017. We added this reference to the final text.
- Did you find any minor/major adverse side effects with these diet during the studies period? If any author has to discuss in the discussion section.
No adverse effect of maltodextrin was observed. We added to the beginning of a result section.
Results
- There was a high probability of assessing glucose tolerance and insulin sensitivity, as well as beta cell and liver function. It has potential to increase the scientific value of this paper. Why did the author not bother to measure these parameters? If you have samples, I recommend to carry out these tests.
Thank you for a very relevant comment. However, the medical objective of our study was to assess whether we can increase CHO intake in diabetic patients requiring follow-up rehabilitation care in a secondary rehabilitation centre. We were only able to measure glycemia and monitor food intake and tolerance of added carbohydrates (maltodextrin). But we were unable to collect blood samples regularly for other research purposes. However, based on our surprising results, we will provide a study to continue.
Discussion
- Discussion section is the paramount of introduction. However, it sounds okay as per their studies, but it gives the reader the opportunity to raise a question because - It has very limited number of patients (sample size very smaller) - Co-relation with previous studies is very limited. - The limitations of the studies in this section are not clear - Lack of future prospective
Thank you for the valuable comment. As far as we know, we are the first study to show that increased intake of carbohydrates did not negatively change blood glucose levels. Moreover, over study duration, the CHO tolerance seemed to improved. In our past study we showed, that incensed intake o glucose did not led to equal need of insulin in diabetic ICU patients, which we mention in the text.
- How is this article more informative than the previously published ones? Justify it.
To our best knowledge this is the first study that shows that increase intake o CHO col be useful for type 2 diabetic patients. We added benefits, limitations and novelty aspect of our study.
Conclusion
- This part looks okay; however, it can be made shorter rather than more descriptive
Thank you for this comment, we made it shorter and more to the point.
The answers to the reviewer 2:
Title: Should carbohydrate intake be liberal during enteral nutrition in type 2 diabetic patients? Comments to Authors Although the study looks interesting, there are issues with the following findings:
Introduction
- The authors should clarify the novelty of this article in the ‘Introduction’ and ‘Conclusion’ section.
We added a sentence to the text to specify strengths and novelty of this work in the end of discussion section. Please see below changes in text.
- Is there any evidence of in vitro or animal model studies? If this is the case, you must discuss it in the introduction and discussion sections. If not, how did you get involved in human research?
Yes, it was documented using in vitro and animal studies previously. Well summarized in review Campbell 2017. We added this reference to the final text.
- Did you find any minor/major adverse side effects with these diet during the studies period? If any author has to discuss in the discussion section.
No adverse effect of maltodextrin was observed. We added to the beginning of a result section.
Results
- There was a high probability of assessing glucose tolerance and insulin sensitivity, as well as beta cell and liver function. It has potential to increase the scientific value of this paper. Why did the author not bother to measure these parameters? If you have samples, I recommend to carry out these tests.
Thank you for a very relevant comment. However, the medical objective of our study was to assess whether we can increase CHO intake in diabetic patients requiring follow-up rehabilitation care in a secondary rehabilitation centre. We were only able to measure glycemia and monitor food intake and tolerance of added carbohydrates (maltodextrin). But we were unable to collect blood samples regularly for other research purposes. However, based on our surprising results, we will provide a study to continue.
Discussion
- Discussion section is the paramount of introduction. However, it sounds okay as per their studies, but it gives the reader the opportunity to raise a question because - It has very limited number of patients (sample size very smaller) - Co-relation with previous studies is very limited. - The limitations of the studies in this section are not clear - Lack of future prospective
Thank you for the valuable comment. As far as we know, we are the first study to show that increased intake of carbohydrates did not negatively change blood glucose levels. Moreover, over study duration, the CHO tolerance seemed to improved. In our past study we showed, that incensed intake o glucose did not led to equal need of insulin in diabetic ICU patients, which we mention in the text.
- How is this article more informative than the previously published ones? Justify it.
To our best knowledge this is the first study that shows that increase intake o CHO col be useful for type 2 diabetic patients. We added benefits, limitations and novelty aspect of our study.
Conclusion
- This part looks okay; however, it can be made shorter rather than more descriptive
Thank you for this comment, we made it shorter and more to the point.
The answers to the reviewer 2:
Title: Should carbohydrate intake be liberal during enteral nutrition in type 2 diabetic patients? Comments to Authors Although the study looks interesting, there are issues with the following findings:
Introduction
- The authors should clarify the novelty of this article in the ‘Introduction’ and ‘Conclusion’ section.
We added a sentence to the text to specify strengths and novelty of this work in the end of discussion section. Please see below changes in text.
- Is there any evidence of in vitro or animal model studies? If this is the case, you must discuss it in the introduction and discussion sections. If not, how did you get involved in human research?
Yes, it was documented using in vitro and animal studies previously. Well summarized in review Campbell 2017. We added this reference to the final text.
- Did you find any minor/major adverse side effects with these diet during the studies period? If any author has to discuss in the discussion section.
No adverse effect of maltodextrin was observed. We added to the beginning of a result section.
Results
- There was a high probability of assessing glucose tolerance and insulin sensitivity, as well as beta cell and liver function. It has potential to increase the scientific value of this paper. Why did the author not bother to measure these parameters? If you have samples, I recommend to carry out these tests.
Thank you for a very relevant comment. However, the medical objective of our study was to assess whether we can increase CHO intake in diabetic patients requiring follow-up rehabilitation care in a secondary rehabilitation centre. We were only able to measure glycemia and monitor food intake and tolerance of added carbohydrates (maltodextrin). But we were unable to collect blood samples regularly for other research purposes. However, based on our surprising results, we will provide a study to continue.
Discussion
- Discussion section is the paramount of introduction. However, it sounds okay as per their studies, but it gives the reader the opportunity to raise a question because - It has very limited number of patients (sample size very smaller) - Co-relation with previous studies is very limited. - The limitations of the studies in this section are not clear - Lack of future prospective
Thank you for the valuable comment. As far as we know, we are the first study to show that increased intake of carbohydrates did not negatively change blood glucose levels. Moreover, over study duration, the CHO tolerance seemed to improved. In our past study we showed, that incensed intake o glucose did not led to equal need of insulin in diabetic ICU patients, which we mention in the text.
- How is this article more informative than the previously published ones? Justify it.
To our best knowledge this is the first study that shows that increase intake o CHO col be useful for type 2 diabetic patients. We added benefits, limitations and novelty aspect of our study.
Conclusion
- This part looks okay; however, it can be made shorter rather than more descriptive
Thank you for this comment, we made it shorter and more to the point.
The answers to the reviewer 2:
Title: Should carbohydrate intake be liberal during enteral nutrition in type 2 diabetic patients? Comments to Authors Although the study looks interesting, there are issues with the following findings:
Introduction
- The authors should clarify the novelty of this article in the ‘Introduction’ and ‘Conclusion’ section.
We added a sentence to the text to specify strengths and novelty of this work in the end of discussion section. Please see below changes in text.
- Is there any evidence of in vitro or animal model studies? If this is the case, you must discuss it in the introduction and discussion sections. If not, how did you get involved in human research?
Yes, it was documented using in vitro and animal studies previously. Well summarized in review Campbell 2017. We added this reference to the final text.
- Did you find any minor/major adverse side effects with these diet during the studies period? If any author has to discuss in the discussion section.
No adverse effect of maltodextrin was observed. We added to the beginning of a result section.
Results
- There was a high probability of assessing glucose tolerance and insulin sensitivity, as well as beta cell and liver function. It has potential to increase the scientific value of this paper. Why did the author not bother to measure these parameters? If you have samples, I recommend to carry out these tests.
Thank you for a very relevant comment. However, the medical objective of our study was to assess whether we can increase CHO intake in diabetic patients requiring follow-up rehabilitation care in a secondary rehabilitation centre. We were only able to measure glycemia and monitor food intake and tolerance of added carbohydrates (maltodextrin). But we were unable to collect blood samples regularly for other research purposes. However, based on our surprising results, we will provide a study to continue.
Discussion
- Discussion section is the paramount of introduction. However, it sounds okay as per their studies, but it gives the reader the opportunity to raise a question because - It has very limited number of patients (sample size very smaller) - Co-relation with previous studies is very limited. - The limitations of the studies in this section are not clear - Lack of future prospective
Thank you for the valuable comment. As far as we know, we are the first study to show that increased intake of carbohydrates did not negatively change blood glucose levels. Moreover, over study duration, the CHO tolerance seemed to improved. In our past study we showed, that incensed intake o glucose did not led to equal need of insulin in diabetic ICU patients, which we mention in the text.
- How is this article more informative than the previously published ones? Justify it.
To our best knowledge this is the first study that shows that increase intake o CHO col be useful for type 2 diabetic patients. We added benefits, limitations and novelty aspect of our study.
Conclusion
- This part looks okay; however, it can be made shorter rather than more descriptive
Thank you for this comment, we made it shorter and more to the point.
The answers to the reviewer 2:
Title: Should carbohydrate intake be liberal during enteral nutrition in type 2 diabetic patients? Comments to Authors Although the study looks interesting, there are issues with the following findings:
Introduction
- The authors should clarify the novelty of this article in the ‘Introduction’ and ‘Conclusion’ section.
We added a sentence to the text to specify strengths and novelty of this work in the end of discussion section. Please see below changes in text.
- Is there any evidence of in vitro or animal model studies? If this is the case, you must discuss it in the introduction and discussion sections. If not, how did you get involved in human research?
Yes, it was documented using in vitro and animal studies previously. Well summarized in review Campbell 2017. We added this reference to the final text.
- Did you find any minor/major adverse side effects with these diet during the studies period? If any author has to discuss in the discussion section.
No adverse effect of maltodextrin was observed. We added to the beginning of a result section.
Results
- There was a high probability of assessing glucose tolerance and insulin sensitivity, as well as beta cell and liver function. It has potential to increase the scientific value of this paper. Why did the author not bother to measure these parameters? If you have samples, I recommend to carry out these tests.
Thank you for a very relevant comment. However, the medical objective of our study was to assess whether we can increase CHO intake in diabetic patients requiring follow-up rehabilitation care in a secondary rehabilitation centre. We were only able to measure glycemia and monitor food intake and tolerance of added carbohydrates (maltodextrin). But we were unable to collect blood samples regularly for other research purposes. However, based on our surprising results, we will provide a study to continue.
Discussion
- Discussion section is the paramount of introduction. However, it sounds okay as per their studies, but it gives the reader the opportunity to raise a question because - It has very limited number of patients (sample size very smaller) - Co-relation with previous studies is very limited. - The limitations of the studies in this section are not clear - Lack of future prospective
Thank you for the valuable comment. As far as we know, we are the first study to show that increased intake of carbohydrates did not negatively change blood glucose levels. Moreover, over study duration, the CHO tolerance seemed to improved. In our past study we showed, that incensed intake o glucose did not led to equal need of insulin in diabetic ICU patients, which we mention in the text.
- How is this article more informative than the previously published ones? Justify it.
To our best knowledge this is the first study that shows that increase intake o CHO col be useful for type 2 diabetic patients. We added benefits, limitations and novelty aspect of our study.
Conclusion
- This part looks okay; however, it can be made shorter rather than more descriptive
Thank you for this comment, we made it shorter and more to the point.
Reviewer 3 Report
The current manuscript investigates the effect of dietary supplementation with a rapidly digestible CHO (maltodextrin) in enteral nutrition to patients in a hospital setting. Patients included those diagnosed with T2D compared to non-diabetic patients to determine glucose responses to the 2 diets in both groups over a 7 day period in a crossover experimental design. The results indicate that the CHO supplement was well tolerated in both groups and that shifts in glycemia with the supplemented diet were acute and transient. The glucose levels in the supplemented group reverted back to values observed before supplementation onset when given 5-7 days to adapt to the new level CHO intake. The authors interpret these results to indicate that provision of supplementary CHO in the form of maltodextrin is safe in a hospital setting for T2D patients.
The authors are careful to present their recommendations in conservative language in that they assert that the CHO supplementation is safe and well tolerated in enteral hospital nutrition. This interpretation appears to be justified by the results, insofar as the data reported. However, some issues should be carefully considered in contextualizing these results. For example, the authors discuss insulin resistance extensively, pointing out that IR can occur in states of stress and inflammation, also during growth periods such as puberty. The authors do not report any data on plasma insulin levels at any point during the study. It is critical to consider not only glucose but insulin levels in conjunction with each other in order to derive conclusions about the metabolic effects of the diet. After all, if a diet does not result in frank hyperglycemia yet causes a significant rise in insulin secretion for the same plasma glucose level there has been a state of relative hyperinsulinemia induced by the diet. The fundamental biochemical lesion in T2D is hyperinsulinemia and insulin resistance and not hyperglycemia per se, which is a consequence of the failure of insulin to properly induce its normal set of cellular responses in sensitive tissues. The authors mention the pentose phosphate pathway in their discussion, which is an interesting focus. However, the concept of metabolic inflexibility in T2D, whereby cells do not readily switch between substrates (fat and CHO) due to mitochondrial overload when presented with an excessive influx of both indicate that mitochondria function best when synthesizing acetyl-CoA from one substrate at a time rather than being inundated with CHO and lipid fuels concurrently (see Muioio, D. 2014). Without data on plasma insulin in these patients, we cannot conclude securely that no adverse metabolic effects were induced by the CHO load from maltodextrin. Perhaps the patients (particularly those with T2D) compensated for the maltodextrin load by hyper-secretion of insulin, in which case the diet can hardly be assumed to be benign. An acknowledgement of the concept of insulin resistance as a protective response to CHO overload in which glycogen stores in insulin sensitive tissues is replete in the presence of abundant influx of dietary CHO would provide more context. In this context, insulin resistance is an overflow phenomenon in which glycogen storing tissues such as liver and skeletal muscle must become insulin resistant in order to be protected from osmotic pressure that would be caused by excessive storage of CHO in the cell, thereby threatening cell lysis.
The writing style and language is overall quite good and only a thorough proofreading for minor grammatical errors is required.
Author Response
The answers to the reviewer 3:
Comments and Suggestions for Authors
The current manuscript investigates the effect of dietary supplementation with a rapidly digestible CHO (maltodextrin) in enteral nutrition to patients in a hospital setting. Patients included those diagnosed with T2D compared to non-diabetic patients to determine glucose responses to the 2 diets in both groups over a 7-day period in a crossover experimental design. The results indicate that the CHO supplement was well tolerated in both groups and that shifts in glycemia with the supplemented diet were acute and transient. The glucose levels in the supplemented group reverted back to values observed before supplementation onset when given 5-7 days to adapt to the new level CHO intake. The authors interpret these results to indicate that provision of supplementary CHO in the form of maltodextrin is safe in a hospital setting for T2D patients.
The authors are careful to present their recommendations in conservative language in that they assert that the CHO supplementation is safe and well tolerated in enteral hospital nutrition. This interpretation appears to be justified by the results, insofar as the data reported. However, some issues should be carefully considered in contextualizing these results.
For example, the authors discuss insulin resistance extensively, pointing out that IR can occur in states of stress and inflammation, also during growth periods such as puberty. The authors do not report any data on plasma insulin levels at any point during the study. It is critical to consider not only glucose but insulin levels in conjunction with each other in order to derive conclusions about the metabolic effects of the diet. After all, if a diet does not result in frank hyperglycaemia yet causes a significant rise in insulin secretion for the same plasma glucose level there has been a state of relative hyperinsulinemia induced by the diet. The fundamental biochemical lesion in T2D is hyperinsulinemia and insulin resistance and not hyperglycaemia per se, which is a consequence of the failure of insulin to properly induce its normal set of cellular responses in sensitive tissues
Thank you for this comment. We agree with reviewer that without plasma inulin the diagnosis of insulin resistance is difficult. We see it as one of the limitations of this study that no direct parameters of insulin resistance were measured. The determination of plasma insulin and possibly C-peptide is very important for the correct interpretation of our results. However, the purpose of our current study was to evaluate the effect of higher CHO intake on plasma glucose measured over 24 hours in diabetics who need follow-up rehabilitation care after an acute illness in order to improve physical activity. The study was organized in a secondary rehabilitation centre. At this workplace, we could only monitor 24-hour glycemia and food intake, including tolerance of added carbohydrates (maltodextrin). However, we could not regularly collect blood samples for other research purposes. Since our results were surprising, we are planning another study in which the level of insulin and C peptide will be monitored. The aim of explanation of insulin resistance was to inform the reader that insulin resistance. We do not share the same opinion as the reviewer that insulin resistance is the consequence of the failure of insulin to properly induce its normal set of cellular responses in sensitive tissues. We see it rather as an important mechanism both in inflammation and physiologic conditions such as growth, development and puberty, which we previously described and published in several manuscripts.
- The authors mention the pentose phosphate pathway in their discussion, which is an interesting focus. However, the concept of metabolic inflexibility in T2D, whereby cells do not readily switch between substrates (fat and CHO) due to mitochondrial overload when presented with an excessive influx of both indicate that mitochondria function best when synthesizing acetyl-CoA from one substrate at a time rather than being inundated with CHO and lipid fuels concurrently (see Muioio, D. 2014).
Without data on plasma insulin in these patients, we cannot conclude securely that no adverse metabolic effects were induced by the CHO load from maltodextrin. Perhaps the patients (particularly those with T2D) compensated for the maltodextrin load by hyper-secretion of insulin, in which case the diet can hardly be assumed to be benign.
An acknowledgement of the concept of insulin resistance as a protective response to CHO overload in which glycogen stores in insulin sensitive tissues is replete in the presence of abundant influx of dietary CHO would provide more context. In this context, insulin resistance is an overflow phenomenon in which glycogen storing tissues such as liver and skeletal muscle must become insulin resistant in order to be protected from osmotic pressure that would be caused by excessive storage of CHO in the cell, thereby threatening cell lysis.
Thank you for the comment. Of course, metabolic inflexibility is a phenomenon that is explained mainly by hormonal changes, but also by mitochondrial disorders. The PPP concept is a relatively new idea that postulates the "utility" of insulin resistance and has been published in more detail in our recent articles (Ref. 9 and 10). Therefore, we focused more on explaining PPP in this section.
The writing style and language is overall quite good and only a thorough proofreading for minor grammatical errors is required.
Changes in text:
Lines 11 – 14 changed to:
Abstract: Carbohydrate (CHO) intake in oral and enteral nutrition is regularly reduced in nutritional support of older patients due to high prevalence of diabetes (usually type 2 - T2DM) in this age group. However, CHO shortage can lead to the lack of building blocks necessary for tissue regeneration and other anabolic processes…..
Line 30:
…patients during oral and enteral…
Line 63:
T2DM development
Lines 79 – 82:
A higher anabolic effect is especially important for patients in whom we expect wound healing or an anabolic course of rehabilitation treatment. The result of the study could be useful for the development of new formulas and products for enteral nutrition and tube feeding, with the aim of increasing their anabolic potential. Therefore the aim…..
Lines 89 – 92:
Levit’s Aftercare Centre in Horice is a long-term hospitalization facility specialized for treatment of patients with chronic wounds or elderly patients in need of a long-term rehabilitation and nutritional care who recently suffered and survived acute illness.
Lines 93 – 103:
Patients were hospitalized in order to improve their general condition after an acute illness or to heal chronic wounds. Physical rehabilitation and nutritional care were the main methods of treatment, and 17 patients underwent local treatment of chronic wounds. Inclusion criteria were as follows: sufficient oral food intake, stable condition of patients without any acute health problems (i.e. fever, vomiting, severe pain, unconsciousness, decompensated diabetes), signed informed consent. Exclusion criteria were: malnutrition with need of artificial nutrition supplements, instability due to acute health issues (see above), terminal stage of disease in and palliative care and refusal to enroll or continue in the study. Brief characterization of patients included in our study is depicted in Supplementary table 1. During the first out of two weeks of this longitudinal crossover study, patients received either a standard institutional diet
Lines 105 – 109:
The composition of both diets were according to the standard protocols at the Levit’s Af-tercare Centre based on recommendations for hospital dietary system in rehabilitation centers in Czech Republic35. Maximum daily carbohydrate intake in the standard diet for non-diabetics was 350 g (57 % of total energy intake) and 250 g (44 % of total energy in-take) for T2DM patients.
Line 115:
this longitudinal crossover
Lines 136 – 140:
We did not observe any major or even minor adverse side effects of maltodextrin supple-mentation throughout the study. The administration of oral antidiabetic drugs or the ad-
ministration of insulin itself was not changed in any patient with T2DM after the addition of maltodextrin to the diet. The energy and CHO intake were …..
Lines 144 – 146:
According to our data, T2DM patients were obese and controls were overweight. We do not have an explanation for this based on our results, but we assume this is perhaps due to well known link between obesity and T2DM.
Lines 203 – 229:
Every disease represents a potential loss of skeletal muscle due to inflammation, inactivity and reduced nutritional intake. The elderly population, where physical activity is limited, usually suffers the most from various acute and chronic diseases with subsequent loss of muscle tissue. Early physiotherapy, rehabilitation and most importantly adequate nutrition are essential for the body-wide anabolic response of the organism3. During growth and regeneration, our dietary substrates must cover both increased energy expenditure and anabolic processes that are needed to improve physical activity and self-sufficiency. Not only must a sufficient total energy intake be supplied, but specific nutritional substrates are also necessary for tissue regeneration. Carbohydrates are important substrates for the initiation and progression of synthetic and anabolic events with subsequent tissue regeneration. This is related to the fact that glucose and its metabolites are necessary building blocks for other molecules and macromolecules, and participate in various metabolic pathways including NADPH production9. Glucose is used for the synthesis of other molecules, which have a fundamental structural and functional importance such as amino acids, nucleic acids and structural molecules such as chondroitin sulphate13, dermatan sulphate, keratan sulphate, and heparan sulphate; or protein glycosylation14. Glycocalyx of the cell membrane14 is vital for cell physiology, such as adhesion, migration, and cell-to-cell communication, or glycosylated intracellular proteins that are associated with cell membrane translocation and exocytosis15. Moreover, the supply of glucose is essential for immune processes and prevention of oxidative stress through the increase of the intracellular pool of NADPH16. NADPH is generated in pentose phosphate pathway (PPP), which uses glucose-6-phosphate17, the main metabolite of glucose. NADPH is used in oxidative killing of bacteria via NADPH oxidase18,19 and is also important for the regulation of immune responses18-21. PPP also generates ribose-5-P, an important precursor for the synthesis of nucleotides or ATP. Precisely for these reasons, glucose is one of the most important nutritional substrates for anabolism9,10.
Lines 230 – 233:
The results of our study show that increased carbohydrate intake did not significantly worsen diabetes compensation in elderly patients with type 2 diabetes, even though the administration of oral antidiabetic drugs and/or insulin did not change during the study.
Lines 276 – 290:
Limitation of this study is that we did not measure directly the insulin sensitivity by glucose/insulin clamp, but we estimated the insulin resistance from fasting and postprandial glycemic curve. Another limitation is, that we did not measure physical parameters which would be interesting to assess, especially in patients undergoing a physiotherapy in order to increase their physical fitness. On the other hand, during preparation of this study, we did not assume a major effect just after one week of high glucose oral feeding. A big benefit of this study is that we were able to see a longitudinal glycemia at rest and postprandial after each meal
in closely controlled environment on human subjects. Whereas in animal studies performed previously34 researchers were able to describe more accurately insulin resistance, their conclusions are always based on animal (mostly rodent) metabolism. The benefit of this study is that we had stable patients with no acute disease. Moreover, we were able to measure portions which were eaten by our participants and closely monitor their dietary intake. Our data were not dependent on self-reporting which usually creates a methodological error. We also confirmed our method and use these sensors in a way which was a novelty for authors to the day of performing this study.
Added references:
Campbell GJ, Senior AM, Bell-Anderson KS. Metabolic Effects of High Glycaemic Index Diets: A Systematic Review and MetaAnalysis of Feeding Studies in Mice and Rats. Nutrients 2017;9(7):646.
Brázdová Z, Fiala J, Bauerová J, Hrubá D. Dietary guidelines in the Czech Republic. I.: Theoretical background and development. Cent Eur J Public Health. 2000 Aug;8(3):186-90
Forouhi NG, Misra A, Mohan V, Taylor R, Yancy W. Dietary and nutritional approaches for prevention and management of type 2 diabetes. BMJ. 2018 Jun 13;361:k2234.
Round 2
Reviewer 1 Report
The manuscript was revised as requested. Comments were made on the important points of criticism and integrated into the text accordingly. The additional information was provided to my satisfaction. Many thanks to the authors.
Reviewer 2 Report
The English language throughout the manuscript needs to be improved.